# Discovery, total syntheses and potent anti-inflammatory activity of pyrrolinone-fused benzoazepine alkaloids Asperazepanones A and B from *Aspergillus candidus*

Li Xu[1,2,5], Feng-Wei Guo[1,2,5], Xue-Qing Zhang[1,5], Tian-Yi Zhou[1], Chao-Jie Wang[1], Mei-Yan Wei[1,3], Yu-Cheng Gu [4], Chang-Yun Wang[1,2] & Chang-Lun Shao [1,2✉]

Natural products are well established as an important resource and play an important role in drug discovery. Here, two pyrrolinone-fused benzoazepine alkaloids, (+)-asperazepanones A (1) and B (2) with a 6/7/5 ring system, together with the artifact (−)-asperazepanone A (1), were isolated from the coral-derived *Aspergillus candidus* fungus. Their structures including absolute configurations were elucidated by extensive spectroscopic methods, single crystal X-ray diffraction, and ECD calculations. Furthermore, total syntheses of (±)-1 and (±)-2 have been achieved starting from the commercially L-aspartic acid diethyl ester hydrochloride and monoethyl malonate in 7 and 8 steps, respectively. The key step in the syntheses was an intramolecular Friedel-Crafts reaction to build the unique tricyclic skeleton. Interestingly, (+)-2 not only showed obviously inhibitory activity against NO production, but also inhibited potent LPS-induced expression of TNF-α and IL-6 at the concentration of 0.1 μM. It thus represents a potentially promising lead for anti-inflammatory drug discovery.

[1] Key Laboratory of Marine Drugs, The Ministry of Education of China, School of Medicine and Pharmacy, Ocean University of China, Qingdao 266003, China. [2] Laboratory for Marine Drugs and Bioproducts, Pilot National Laboratory for Marine Science and Technology (Qingdao), Qingdao 266200, China. [3] College of Food Science and Engineering, Ocean University of China, Qingdao 266003, China. [4] Syngenta Jealott's Hill International Research Centre, Bracknell, Berkshire RG42 6EY, UK. [5] These authors contributed equally: Li Xu, Feng-Wei Guo, Xue-Qing Zhang. ✉email: shaochanglun@163.com

nflammation is a natural biological response to infection and injury, directed towards eliminating harmful agents and restoration of the normal function of the affected tissue or organ[1,2]. Inflammatory diseases remain a major health problem worldwide. A large number of nonsteroidal anti-inflammatory drugs (NSAIDs)[3,4] such as aspirin, diclofenac, and indomethacin have emerged in the past few decades, and the development of COXIBs[5,6] has provided some relief to patients suffering from a wide spectrum of inflammatory diseases. However, these drugs often cause cardiovascular side effects which has subdued the medicinal applications of this class of anti-inflammatory drugs[7–10]. Therefore, new anti-inflammatory agents with higher efficacy and lower toxicity are urgently required.

Natural products have proven to be a reliable resource of new pharmaceutical agents[11,12]. Natural alkaloids and their derivatives with vast diversity of structures are a cornerstone of chemotherapeutic regimens worldwide[11,13]. Benzoazepines are a key structural motif in numerous pharmaceuticals and agrochemicals[14–16]. Particularly, a benzo[*b*]azepine moiety is a well-known pharmacophore that is present in several marketed tricyclic antidepressant drugs, including anafranil, imipramine, tienopramine, and amezepine (Fig. 1a)[17]. In addition, this core structure plays key roles in other biological activities including anticonvulsant, anti-HIV-1, and antiallergic properties[18–20].

During our ongoing search for new biologically active constituents from marine fungi, a series of bioactive natural products with antifungal, antimalarial, and antiviral activities have been discovered[21–25]. For example, plentifully functionalized *p*-terphenyl metabolites with attractive pharmacological activities were isolated from the gorgonian coral-derived *Aspergillus candidus* (CHNSCLM-0393) fungus[25]. The chemical constituents of the fungus were further studied under the guidance of LC-MS/MS-Based Molecular Networking and HPLC-UV-DAD, and two new benzoazepine alkaloids with an unprecedented tricyclic 6/7/5 skeleton, asperazepanones

A (**1**) and B (**2**), were discovered (Fig. 1b). Herein, the isolation, structure elucidation, total syntheses, and anti-inflammatory activity of **1** and **2** are described.

## Results

**Structure elucidation of (±)-asperazepanone A (1) and (+)-asperazepanone B (2).** Asperazepanone A (**1**) was obtained as a yellow amorphous solid. Its molecular formula of $C_{12}H_{10}N_2O_3$ (nine degrees of unsaturation) was determined by the positive HRESIMS (m/z 231.0763 [M + H] +, calcd. 231.0764). The IR spectrum in KBr clearly suggested the presence of carbonyl group (1624 cm$^{-1}$). The $^1$H NMR spectrum (see Supplementary Table S1) revealed these signals for three aromatic proton signals ($\delta_H$ 6.87, t, J = 7.9 Hz; 7.06, dd, J = 7.9, 1.2 Hz; 7.27, dd, J = 7.9, 1.2 Hz), one olefinic proton ($\delta_H$ 5.13, 1H, s), one methylene proton ($\delta_H$ 2.85, dd, J = 17.2, 13.1 Hz; 3.03, dd, J = 17.2, 2.8 Hz), and one methine proton ($\delta_H$ 4.51, dd, J = 13.1, 2.8 Hz). Furthermore, three exchangeable protons at $\delta_H$ 10.46, 9.13, and 7.22 were observed in DMSO-$d_6$. Its $^{13}$C NMR and DEPT spectra exhibited a total of 12 carbon resonances divided into one methylene group, five methines, and six quaternary carbons.

The HSQC spectrum clearly indicated the presence of a keto carbonyl ($\delta_C$ 195.1), and an amide ($\delta_C$ 173.9) in the molecule. The two carbonyl groups, and one olefinic carbon accounted for three degrees of unsaturation, the remaining six degrees of unsaturation indicated the presence of a tricyclic system in the structure of **1**. The consecutive COSY correlation from H-6 to H-8, together with the HMBC correlations from aromatic H-6 ($\delta_H$ 7.06) and H-8 ($\delta_H$ 7.27) to C-4a ($\delta_C$ 129.5), from H-7 ($\delta_H$ 6.87) to C-8a ($\delta_C$ 125.5), and from hydroxy group signal at $\delta_H$ 11.49 (s) to C-5 ($\delta_C$ 147.2) and C-4a, indicating the presence of a disubstituted phenol ring A. The adjacent COSY correlation from H-10 ($\delta_H$ 2.85, dd, J = 17.2, 13.1 Hz; 3.03, dd, J = 17.2, 2.8 Hz) to H-10a ($\delta$H 4.51, dd, J = 13.1, 2.8 Hz) indicated the presence of –CH$_2$CH– unit. The ring C was constructed on the basis of the following correlations in the COSY spectrum of H-1 ($\delta_H$ 7.22) to H-10a and H-3 ($\delta_H$ 5.13), and the HMBC correlations from H-1 to C-10a ($\delta_C$ 50.2), C-3 ($\delta_C$ 92.9), and C-3a ($\delta_C$ 162.8), H-3 to C-10a, and C-2 ($\delta_C$ 173.9), and H-10a to C-3a. The correlation of H-4 ($\delta_H$ 9.13) to H-10a and H-1, and HMBC correlations from H-4 to C-5 ($\delta_C$ 147.2), C-8a ($\delta_C$ 36.5), and C-10a, H-8 to C-9 ($\delta_C$ 195.1), C-6 ($\delta_C$ 118.6) and C-4a ($\delta_C$ 129.5), and H-10 to C-9, C-10a ($\delta_C$ 36.5), and C-3a, revealed the presence of a seven-membered ring B that was fused to the rings A and C. Thus, an unexpected pyrrolinone fused benzoazepine alkaloid with a tricyclic 6/7/5 skeleton was elucidated as shown in Fig. 2a.

Asperazepanone B (**2**) was obtained as a pale yellow crystal with a molecular formula of $C_{13}H_{12}N_2O_3$ determined by HRESIMS (m/z 245.0917 [M + H] +, calcd. 245.0921). Both the $^1$H and $^{13}$C NMR data of **2** were closely related to those of **1**, with the exception of the presence of an additional methyl ($\delta_H$ 3.16, $\delta_C$ 40.3). The position of the methyl was confirmed by the HMBC spectrum, which showed correlations from H$_3$-4 to C-8a ($\delta_C$ 129.4), and C-3a ($\delta_C$ 166.4), and we could determine the planar structure of **2**.

To unambiguously assign the unprecedented structure and absolute configuration, a single X-ray diffraction study was performed. However, compound **1** was initially isolated as an amorphous solid and it was difficult to get a suitable crystal for X-ray analysis, but after many attempts, X-ray quality crystals were obtained by slow evaporation of a solution of **1** in MeOH (Fig. 2b). Further analysis of the X-ray data revealed that **1** possessed a centrosymmetric space group $P2_1/c$, indicating its racemic nature (Supplementary Data 1). Fortunately, a recrystallization experiment

**Fig. 1 Representative examples of benzoazepines. a** Commercial benzoazepine drugs. **b** Chemical structures of (±)-**1** and (+)-**2**.

also furnished a suitable crystal of **2** (Fig. 2b). To our surprise, Cu Kα X-ray crystallographic analysis confirmed the structure of **2** with absolute configuration of 10a*S* with a Flack parameter of −0.1(3) (Supplementary Data 2). And as shown in Fig. 3, the calculated ECD curve of **2** is also in agreement with the experimental one.

In order to clarify the racemic mixture of **1**, HPLC analysis of **1** on a chiral column was carried out. Two distinct chromatographic peaks with a ratio of 3:2 were found (Fig. 4). After chiral HPLC separation, compounds (+)-**1** and (−)-**1** were obtained. To determine the absolute configurations of (+)-**1** and (−)-**1**, the experimental CD spectra and the calculated ones using the time-dependent DFT method of each enantiomer were compared. The results showed that the ECD curve of 10a*S*-**1** was similar to that of (+)-**1** and the ECD curve of 10a*R*-**1** was similar to that of (−)-**1** (Fig. 3). On the basis of the above evidence, the absolute configurations of (+)-**1** and (−)-**1** were established to be 10a*S* and 10a*R*, respectively.

**Differentiation and epimerization property of 1 and 2.** With a systematic investigation on the structure elucidation of **1** and **2** completed, another question arose: why was compound **1** isolated as a racemate and compound **2** isolated as an optically active? Therefore, we wanted to determine whether the racemates of **1** were naturally occurring or obtained as artifacts during the isolation process. After 10 days of short-term fermentation, **1** was quickly separated by RP-HPLC and analyzed by chiral HPLC in 1 day. The results showed that (+)-**1** was detected as the major one (95%; see Supplementary Fig. S7). Therefore, (+)-**1** should be regarded as a true natural product, while (−)-**1** is an artifact.

To explore the factors of the transformation from (+)-**1** to (−)-**1**, the following experiments were carried out (see Supplementary Figs. S8 and S9). To our surprise, when (+)-**1** was treated with 0.01% NaOH, 10% acetic acid, or 2% trifluoroacetic acid (TFA) in MeOH at room temperature, respectively, (+)-**1** could be partially transformed into (−)-**1** accompanied with its dehydrogenation product **3** (Fig. 5a). As we expected, **2** was relatively stable under the above conditions (see Supplementary Fig. S10). After eight months of fermentation, the pH value of the medium was close to 9, so we deduced that the racemization of (+)-**1** was caused by long-time fermentation. Next, we deduced the mechanism of racemic reaction of **1** (Fig. 5b). At the beginning, enamine-imino tautomerism[26] of (+)-**1** might provide the intermediate A, which immediately underwent a second imino-enamine tautomerism to produce the intermediate B. Finally, it further underwent a racemization reaction to deliver (−)-**1**. As for **2**, due to the lack of active hydrogen on the nitrogen atom, the corresponding intermediate A cannot be formed.

**Plausible biosynthetic pathway to 1 and 2.** Structurally, compounds **1** and **2** are unprecedented benzoazepine alkaloids with a tricyclic 6/7/5 skeleton, which is the first examples of benzoazepine alkaloid, generating by the fusion of a benzoazepine scaffold with a pyrrolinone moiety. Plausible biogenetic pathways for **1** and **2** are postulated in Fig. 6. As shown, the unique biogenetic route could be plausibly traced back to L-tryptophan.

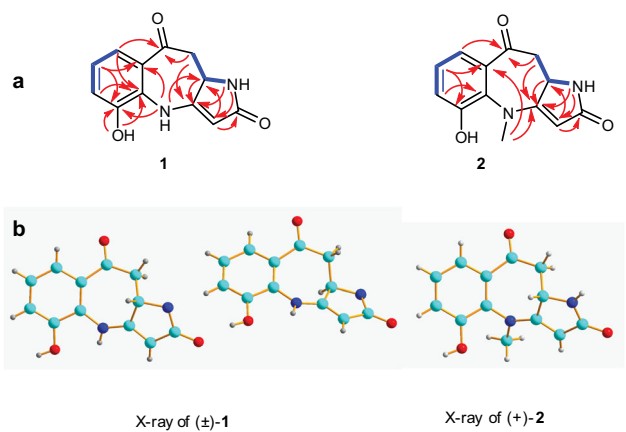

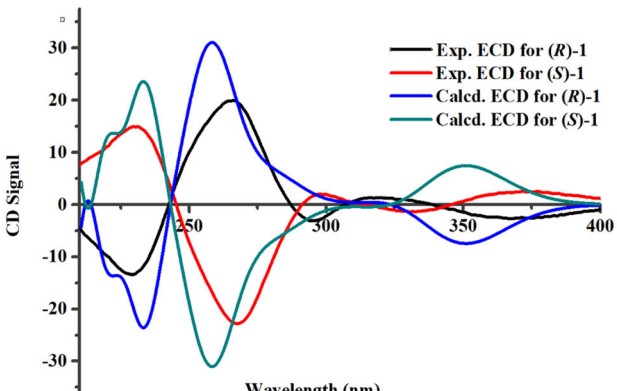

**Fig. 2 Structure elucidation. a** COSY and key HMBC correlations of **1** and **2**. **b** X-ray structures of (±)-**1** and (+)-**2**.

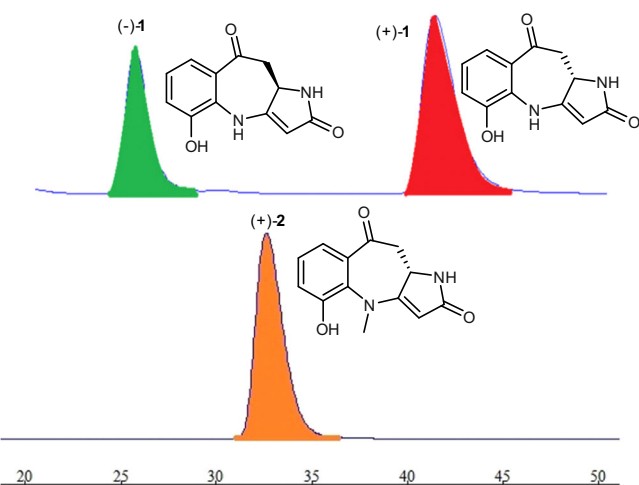

**Fig. 4 Chiral-phase HPLC analysis.** Chiral-phase HPLC analysis of (±)-**1** and (+)-**2**.

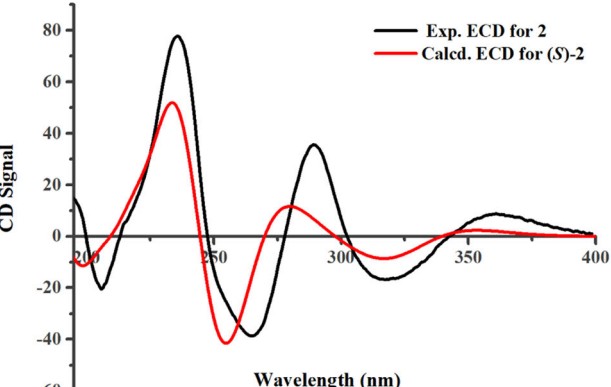

**Fig. 3 Comparison between calculated and experimental ECD spectra.** Experimental ECD spectra and calculated ECD spectra of (±)-**1** and (+)-**2**.

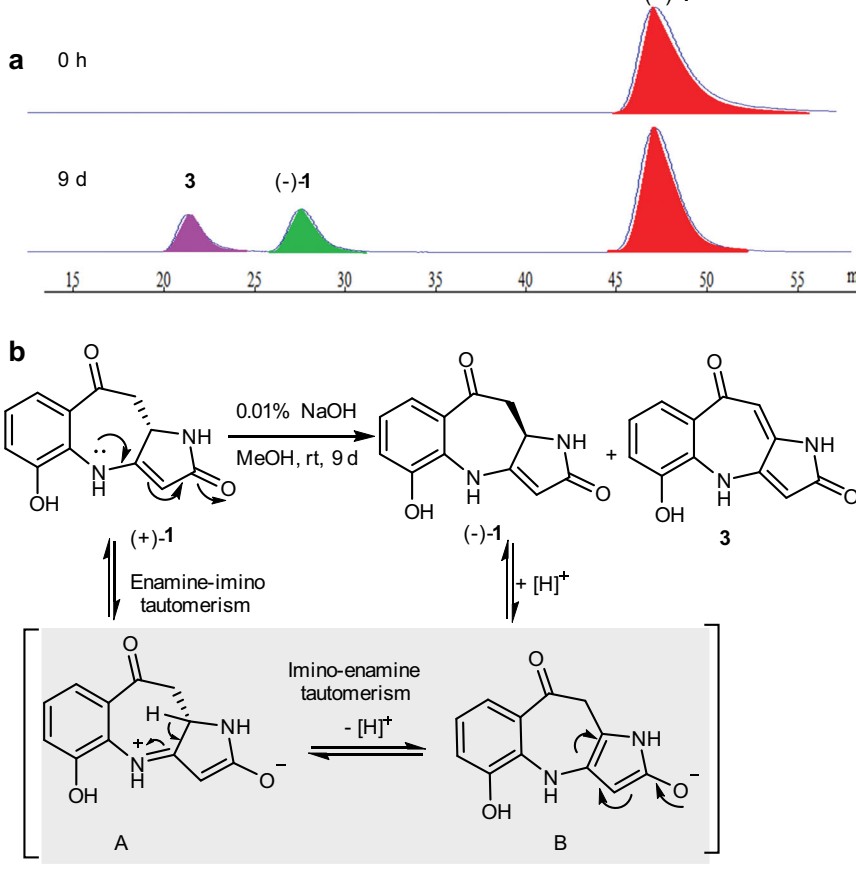

**Fig. 5 Epimerization property. a** The transformation from (+)-10a*S*-**1** to (−)-10a*R*-**1** and **3** in MeOH with 0.01% NaOH. **b** The plausible mechanism of racemization for **1**.

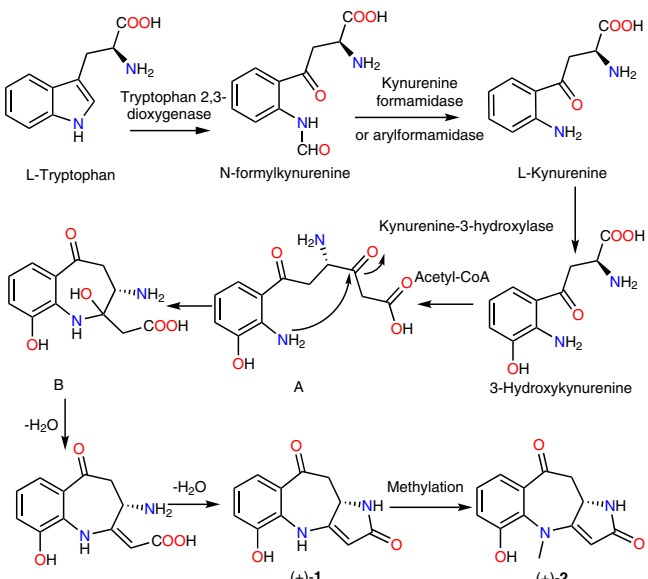

**Fig. 6 Biogenetic pathway.** Plausible biosynthetic pathway to **1** and **2**.

L-Tryptophan has been known as a precursor of 3-hydroxykynurenine[27,28]. Then, 3-hydroxykynurenine reacted with acetyl-CoA to form intermediate A. Further, the intermediate A underwent an intramolecular nucleophilic addition to give the bicyclic intermediate B. Finally, (+)-**1** was produced by the two-step dehydration of intermediate B. (+)-**2** was further obtained by the methylation of (+)-**1**.

**Retrosynthetic analysis of (±)-asperazepanones A (1) and B (2).** Benzoazepines are frequently encountered in pharmaceuticals and exhibit significant biological activities, and thus, they have attracted considerable interest from the synthetic community[29–32]. Therefore, the total syntheses of **1** and **2** were conducted. The retrosynthetic analysis for the total syntheses of **1** and **2** are delineated in Fig. 7. We assumed that **1** and **2** might be assembled from acid **4** through an intramolecular Friedel-Crafts reaction[33]. For the construction of the latter, the nucleophilic addition[34] of the aromatic amine **5** to the 3-carbonyl of tetramic acid **6** was projected as the means to forge the skeleton of **4**. The 3-carbonyl of tetramic acid **6** would be obtained through the decarboxylation of the substituted tetramic acid **7**. Finally, the core of **7** could be constructed via Dieckmann cyclization[35] of amidation intermediate, which could be derived from the commercially available L-aspartic acid diethyl ester hydrochloride (**8**) and monoethyl malonate (**9**).

**Total syntheses of (±)-asperazepanones A (1) and B (2).** As depicted in Fig. 8, our synthetic study began with the preparation of amide **10**. Monoethyl malonate (**9**) reacted with commercially available L-aspartic acid diethyl ester hydrochloride (**8**) in the presence of 1-[3-(dimethylamino)propyl]-3-ethylcarbodiimide hydrochloride (EDCI), diisopropylethylamine, and a catalytic amount of 4-dimethylaminopyridine to provide amide **10** in 90% yield. Next, Dieckmann cyclization of the optical amide **10** delivered the racemic tetramic acid **7** in 70% yield. Not surprisingly, although the Dieckmann cyclization strategy is a convenient way of preparing tetramic acids, it is accompanied by non-negligible epimerization at position C-5[36,37]. The attempt to

**Fig. 7 Synthetic strategy.** Retrosynthesis of asperazepanones A (**1**) and B (**2**).

**Fig. 8 Preparation of fragment 4.** Reagents and conditions: **a** DMAP, DIPEA, EDCI, $CH_2Cl_2$, 0 °C to RT, 12 h, 90%; **b** LiHMDS, THF, −20 °C to RT, 6 h, 70%; **c** TFA/MeCN (1: 10), reflux, 8 h; **d** 5, AcOH/EtOH (1: 10), 70 °C, 12 h, 30% over two steps; **e** LiOH, MeOH/$H_2O$ (1: 1), RT, 2 h, 85%. DMAP 4-dimethylaminopyridine, DIPEA diisopropylethylamine, EDCI 1-(3-dimethylaminopropyl)-3-ethylcarbodiimide hydrochloride, LiHMDS lithium hexamethyldisilazide. TFA trifluoroacetic acid, THF tetrahydrofuran, RT room temperature.

obtain the desired 3-carbonyl of tetramic acid **6** through decarboxylation reaction of **7** was unsuccessful according to the literature procedure[34]. Fortunately, refluxing **7** in a mixture of acetonitrile and TFA resulted in decarboxylation and afforded the 3-carbonyl of tetramic acid **6**.

Without purification, the nucleophilic addition reaction of **6** was then carried out. Gratifyingly, by treating **5** and **6** with acetic acid in ethanol at 70 °C, the adduct **11** was obtained in 30% yield over two steps. Enantioselective synthesis of **11** starting from enantiomerically pure amino acids was not pursued because of reported partial racemization under the cyclization conditions[36]. With the successful synthesis of **11**, we turned our attention to the construction of the unusual 6/7/5 ring system. After hydrolysis of **11**, the resulting acid **4** was ready for the intramolecular Friedel–Crafts reaction.

Intramolecular Friedel−Crafts cyclization was found to be the crucial step in these syntheses. As summarized in Table 1, initial attempts to carry out the intramolecular Friedel−Crafts reaction of **4** using TFAA as cyclization reagent were unsuccessful, only byproduct **12a** was obtained (entry 1). On the basis of the above experiments, different reagents ($SOCl_2$/$AlCl_3$, $BF_3$·$Et_2O$, $POCl_3$, $H_2SO_4$, and PPA) were screened at this stage in different reaction conditions. Our various attempts to obtain the desired **12** through this coupling reaction, employing different solvents and Lewis acid, did not succeed (entries 2–5). Fortunately, upon treatment of **4** with PPA at 80 °C, we were able to successfully isolate the desired **12** in 3% yield (entry 6). The highest yield (22%) of **12** was obtained with PPA at 120 °C for 1 h (entry 8).

With the key intermediate **12** in hand, we now move to the stage for the completion of the total synthesis (Fig. 9). Finally, treatment of **12** with $BBr_3$ in $CH_2Cl_2$ at room temperature furnished (±)-asperazepanone A (**1**) in 74% yield. Subsequent methylation of **12** and $BBr_3$-mediated phenol demethylation under standard conditions furnished (±)-asperazepanone B (**2**) in 68% yield over two steps. The spectral and physical data of fully synthetic **1** and **2**, including $^1H$ NMR, $^{13}C$ NMR, and HRMS data, matched those of naturally occurring **1** and **2** (see

Supplementary Note 2, Tables S2 and S3, Figs. S25 and S26). (±)-Asperazepanone B (**2**) was resolved into the corresponding pure enantiomers (+)-**2** and (−)-**2** by HPLC using a Chiralpak IC chiral-phase column.

**The anti-inflammatory activity of (±)-asperazepanone A (1), (+)- and (−)-asperazepanone B (2).** A macrophage stimulation assay is often used as a validated screening tool for anti-inflammatory compounds[38,39]. Excess nitric oxide (NO) generation is a sign of the inflammatory response[40]. Pro-inflammatory cytokines, such as IL-6 and TNF-α, have been recognized as critical regulators in inflammatory diseases[41] such as rheumatoid arthritis[42], inflammatory bowel disease[43], asthma[44], and diabetic complications[45]. Inhibiting the release of pro-inflammatory cytokines is an important mode of action for anti-inflammatory drugs[46].

In current research, the effects of compounds (±)-**1**, (+)-**2**, and (−)-**2** in vitro in lipopolysaccharide (LPS)-stimulated RAW 264.7 cells were evaluated (Fig. 10). Only (+)-**2** did not affect the cell viability and showed obviously inhibitory activity against nitric oxide (NO) production with inhibition rate of 43 ± 4% at the concentration of 1 μM. However, Other compounds (±)-**1** and (−)-**2** did not show any inhibitory activity. L-NMMA was used as the positive control with the rate of 61 ± 11% at the concentration of 5 μM. To further measure the cytokine release level, pro-inflammatory cytokines TNF-α and IL-6 were detected by ELISA. The levels of TNF-α ($p < 0.0001$) and IL-6 ($p < 0.01$) were significantly decreased with 40 ± 2% and 77 ± 7% inhibition rates, respectively, by (+)-**2** at the concentration of 0.1 μM (Fig. 10)

**Table 1 Optimization of reaction conditions.**

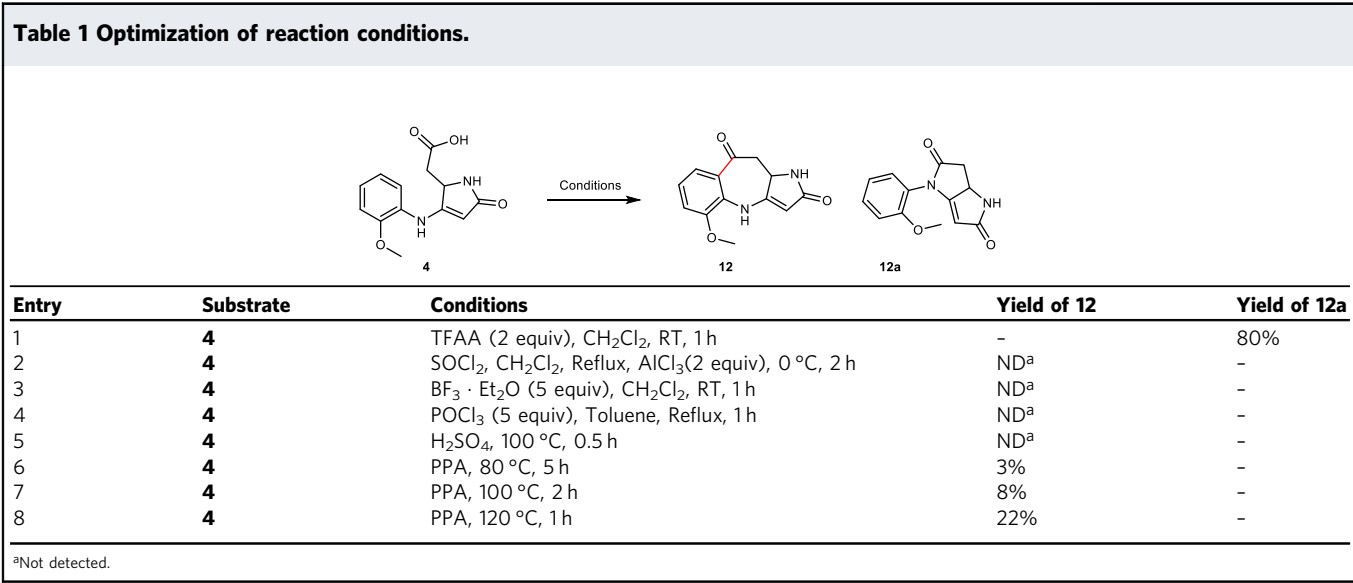

| Entry | Substrate | Conditions | Yield of 12 | Yield of 12a |
|---|---|---|---|---|
| 1 | 4 | TFAA (2 equiv), CH$_2$Cl$_2$, RT, 1 h | – | 80% |
| 2 | 4 | SOCl$_2$, CH$_2$Cl$_2$, Reflux, AlCl$_3$(2 equiv), 0 °C, 2 h | ND[a] | – |
| 3 | 4 | BF$_3$ · Et$_2$O (5 equiv), CH$_2$Cl$_2$, RT, 1 h | ND[a] | – |
| 4 | 4 | POCl$_3$ (5 equiv), Toluene, Reflux, 1 h | ND[a] | – |
| 5 | 4 | H$_2$SO$_4$, 100 °C, 0.5 h | ND[a] | – |
| 6 | 4 | PPA, 80 °C, 5 h | 3% | – |
| 7 | 4 | PPA, 100 °C, 2 h | 8% | – |
| 8 | 4 | PPA, 120 °C, 1 h | 22% | – |

[a]Not detected.

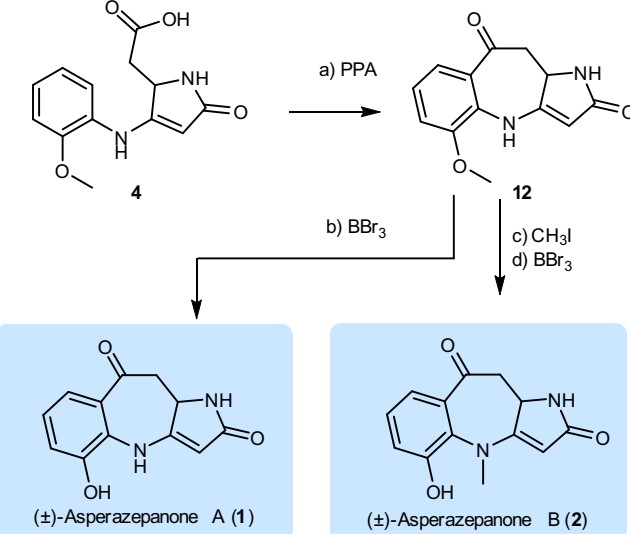

**Fig. 9 Completion of the total syntheses of asperazepanones A and B.**
Reagents and conditions: **a** PPA, 120 °C, 1 h, 22%; **b** BBr$_3$, CH$_2$Cl$_2$, RT, 2 h, 74%; **c** CH$_3$I, Cs$_2$CO$_3$, acetone, RT, 6 h; **d** BBr$_3$, CH$_2$Cl$_2$, RT, 2 h, 68% over two steps. PPA polyphosphoric acid, RT room temperature.

without affecting cell viabilities (See Supplementary Note 3 Fig. S60). Very interestingly, there is obvious difference in the anti-inflammatory activity of each enantiomer of asperazepanone B (**2**). The results revealed that (+)-**2** with *S* configuration exhibited potent inhibitory activity with these two cytokines, while (−)-**2** with *R* configuration was inactive. The preliminary structure-activity relationships indicated that the absolute configuration of (*S*)-**2** plays a key role in the observed anti-inflammatory activity. Moreover, (±)-(**1**) exhibited no inhibitory activity suggesting that the methyl group at the N-4 is also crucial to the anti-inflammatory activity. Our results suggest that the novel pyrrolinone-fused benzoazepine alkaloid, (+)-asperazepanone B (**2**) deserves further investigations as potential anti-inflammatory agents.

## Conclusion

In summary, we have isolated and characterized two novel natural products, (+)-asperazepanones A (**1**) and B (**2**), from the gorgonian derived fungus *Aspergillus candidus*. The artifact

(−)-asperazepanone A (**1**) was confirmed and its transformation mechanism was also studied. Moreover, the total syntheses of (±)-**1** and (±)-**2** featuring intramolecular Friedel-Crafts reaction to construct the tricyclic 6/7/5 architecture were also accomplished. This synthetic process should provide routes for obtaining synthetic analogs. Biological studies revealed that (+)-asperazepanone B (**2**) showed potent anti-inflammatory activity, which confirmed it as promising new lead for developing anti-inflammatory agent. Further structural optimization, biological evaluation, and mechanism of action are ongoing and will be reported in due course.

## Materials and methods

**General**. All reactions were performed under nitrogen or argon in vacuum-dried glassware using dry solvents and standard syringe techniques. Reagents were purchased at the highest commercial quality (>95%) and used without further purification, unless otherwise stated. Anhydrous tetrahydrofuran (THF) was distilled from sodium-benzophenone, dichloromethane (CH$_2$Cl$_2$) was distilled from calcium hydride. Column chromatography was carried out by using silica gel (200–300 mesh). Yields refer to chromatographically, unless otherwise specified. Optical rotations were measured on a JASCO P-1020 digital polarimeter (JASCO Ltd., Tokyo, Japan). ECD spectra were acquired on a JASCO J-715 (JASCO) or Chirascan CD (Applied Photophysics) spectropolarimeter. IR spectra were obtained on a Nicolet Nexus 470 spectro-photometer (Perkin Elmer Ltd., Boston, MA, USA) in KBr discs. NMR spectra were recorded on a JEOL JEM-ECP NMR spectrometer (JEOL Ltd., Tokyo, Japan; 500 MHz for [1]H and 125 MHz for [13]C), using TMS as an internal standard. HRESIMS spectra were performed on a Thermo Scientific LTQ Orbitrap XL spectrometer. Single-crystal data were obtained on an Agilent Gemini Ultra diffractometer (Cu Kα radiation) (Agilent Technologies Inc., California, America). HPLC analysis was performed on a Hitachi L-2000 system (Hitachi Ltd., Tokyo, Japan) using a C18 column [(YMC Co., Ltd., Tokyo, Japan) YMC-Park, ODS-A, 250 × 4.6 mm, S-5 μm, 12 nm, 1.0 mL/min]. Semi-preparative HPLC was performed on a Hitachi L-2000 system (Hitachi Ltd., Tokyo, Japan) using a C18 column [(Eka Ltd., Bohus, Sweden) Kromasil 250 × 10 mm, 5 μm, 2.0 mL/min]. Racemic mixtures were resolved on a Chiralpak IC column (5 μm, 4.6 × 150 mm, hexane−ethanol eluent, 0.6 mL/min) and Chiralpak IA column (5 μm, 4.6 × 250 mm, hexane−ethanol eluent, 0.6 mL/min). Silica gel ((Qingdao Haiyang Chemical Group Co., Qingdao, China; 200–300 mesh), octadecylsilyl silica gel (YMC Co., Ltd., Tokyo, Japan; 45–60 μm), macroporous resin (H&E Co., Ltd., Connecticut, America) and Sephadex LH-20 (GE Ltd., Connecticut, America) were used for column chromatography. Precoated silica gel plates (Yantai Zhifu Chemical Group Co., Yantai, China; G60, F-254) were used for thin layer chromatography.

**Reporting summary**. Further information on research design is available in the Nature Research Reporting Summary linked to this article.

## Data availability

The authors declare that all data supporting the findings of this study are available within the article and its supplementary information files, and from the corresponding

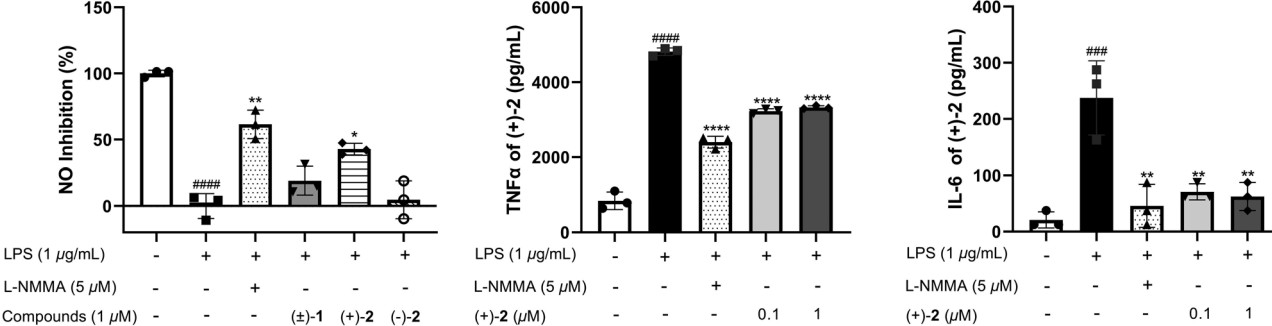

**Fig. 10 Effects of compound on NO, TNF-α and IL-6 production in LPS-induced RAW 264.7 cells.** The cells were treated with the different concentrations and LPS (1 μg/mL) for 24 h. Data are presented as the mean of three experiments ± SD. ###$p < 0.001$, ####$p < 0.0001$ compared to the control group; *$p < 0.05$, **$p < 0.01$, ***$p < 0.001$, ****$p < 0.0001$ compared to the LPS-treated group.

author on request. For [1]H and [13]C NMR spectra of compounds, see Supplementary Note 1 Figs. S11–S24 and Supplementary Note 2 Figs. S46–S59. For the spectroscopic and physical data of compounds, see Supplementary Note 1 Figs. S1–S6 and Supplementary Note 2 Figs. S27–S45. The X-ray crystallographic coordinates for the structure of 1 and 2 have been deposited at the Cambridge Crystallographic Data Center (CCDC), under deposition number 2143832 and 2143833. These data can be obtained free of charge from The Cambridge Crystallographic Data Center via www. ccdc.cam.ac.uk/data_request/cif. Other data are available from the authors upon reasonable request. The CIF files of CCDC 2143832 and CCDC 2143833 are also included as Supplementary Data 1 and 2.

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

## Acknowledgements

We thank Syngenta for the fellowship to Feng-Wei Guo. This work was supported by the Fundamental Research Funds for the Central Universities (No. 202264001), the Program of National Natural Science Foundation of China (Nos. U1706210 and 41322037), AoShan Talents Program Supported by Pilot National Laboratory for Marine Science and Technology (Qingdao) (No. 2015ASTP-ES11), Key Laboratory of Tropical Medicinal Resource Chemistry of Ministry of Education, Hainan Normal University (Nos. RDZH2021003 and RDZH2022002), the Program of Natural Science Foundation of Shandong Province of China (No. JQ201510), and the Taishan Scholars Program, China (No. tsqn20161010).

## Author contributions

C.L.S. supervised and designed the project. L.X., F.W.G., X.Q.Z., and C.J.W. performed chemical experiments. T.Y.Z. contributed to biological experiments. L.X., F.W.G., X.Q.Z., and C.L.S. co-wrote the manuscript. C.L.S., L.X., F.W.G., M.Y.W., Y.C.G., and C.Y.W. discussed the results and assisted in the preparation of this manuscript.

## Competing interests

The authors declare no competing interests.
