## [Peer review file · Communications Chemistry]

Reviewers' comments:

Reviewer #1 (Remarks to the Author):

The manuscript submitted by Shao and coworkers describes isolation, synthesis and biological evaluation of novel natural products, asperazepanones A and B. The authors isolated these natural products from *Aspergillus candidus*. The structures were determined by the standard NMR techniques and X-ray crystallography. They also revealed that asperazepanone A easily racemizes, while asperazepanone B is optically stable. The difference is attributed to the substitution of a methyl group on the nitrogen atom. The authors successfully synthesized these natural products, although the key Friedel-Crafts reaction proceeded only in low yield (compound 4 to 12, 22% yield). Biological evaluation of the natural products was also conducted, revealing that (+)-asperazepanone B showed inhibitory activity against nitric oxide (NO) production and inhibition on LPS-induced expression of TNF- α and IL-6.

The achievement in this manuscript will be informative for readers, because the authors disclosed chemistry and biology of natural products which have an unprecedented core, 6/7/5 diazatricyclic system. On the other hand, the manuscript has critical issues. This reviewer recommends publication of this work in this journal after the issues are properly addressed.

(1) preparation of (-)-asperazepanone B: Although the authors used (-)-asperazepanone B, the enantiomer of the natural product, in the biological evaluation, the method for preparation of (-)-asperazepanone B is not described in the manuscript and Supporting Information. It must be included.

(2) For the synthesis of asperazepanones A and B, the authors used L-aspartic acid dimethyl ester as a substrate, but they mentioned in the manuscript that they synthesized natural products in racemic forms. Evidences that the products were racemic are not provided, and these should be added. In addition, this reviewer recommends the authors to add comments about when racemization occurred.

(3) It is recommended to add the optical rotation values ($[\alpha]_D$) for the optically active compounds.

(4) It is better that the structure of compound 3 appears in the manuscript.

(5) Supporting Information, page 24, line 334: CsCO₃  Cs₂CO₃

Reviewer #2 (Remarks to the Author):

The manuscript „Discovery, Total Syntheses and Potent Anti-inflammatory Activity of Asperazepanones A and B, Two Unique Pyrrolinone-Fused Benzoazepine Alkaloids from *Aspergillus candidus*“ was submitted to Communications Chemistry for publication.

Broad comments:

The study describes the isolation of two novel benzoazepine alkaloids from *Aspergillus* cultures and the total synthesis of both isolated compounds. Moreover, some thoughts on the biosynthesis of the compounds are given and activity studies on NO production as well as on the expression of TNF- α and IL-6 are presented.

All the experiments are well described and the isolation and structure elucidation as well as the synthesis of the compounds is easy to follow. However, there are still some points to be addressed.

Figures: In contrast to the text, which is well written, some of the figures are misleading. In Figure 6, for example, which is showing the enamine-imino-tautomerism, in the starting molecule [(+)-1], the double bonds should move from the keto-oxygen towards the nitrogen. Therefore, the arrows should show in the other direction and one big arrow should then show from the hydrogen to the keto-

oxygen. Likewise, in the next molecule the arrow should not show towards the nitrogen but from the nitrogen and so on. Also, Figure 8 is misleading. Even though, it is a retrosynthetic analysis the fact that e.g. the arrow showing the decarboxylation is heading towards the carboxylated molecule appears somewhat strange. Maybe, a double pointed arrow would be more appropriate. Also for the other depicted reactions.

Biological activity: The authors present effects of the compounds on NO production and on the LPS-induced expression of TNF- α and IL-6, claiming (+)-asperazepanone B as promising lead structure. However, no data on the cytotoxicity of the compound are given, which must not be neglected due to the known toxicity of *Aspergillus* species.

Specific comments:

Lines 58 and 82: The calculated m/z ratios of 1 and 2 should be 231.0770 and 245.0926, respectively.

Line 114: Please write "partially transformed".

Line 158: Please write "Friedel-Crafts" here and in the whole manuscript.

Line 207 ff: Please indicate manufacturer and city and country of origin in the materials and methods section.

Dear Editor:

Thank you very much for your kind supports and for the reviewers' comments concerning our manuscript entitled "Discovery, total syntheses and potent anti-inflammatory activity of asperazepanones A and B, two unique pyrrolinone-fused benzoazepine alkaloids from *Aspergillus candidus*" (manuscript ID COMMSCHEM-22-0131-T). The suggestions and comments are all valuable and very helpful for revising and improving our manuscript. We have studied these carefully and made efforts for the corrections. Revised portions are marked in yellow color in the "Revised Manuscript" and "Supplementary Information" files. The main corrections in the manuscript and reviewers' comments are as follows:

Editor: Please make sure to unambiguously clarify the racemization of the synthetic asperazepanones, provide optical rotation values of the optically active compounds and detail the preparation of (–)-asperazepanone B. Should further experimental data or analysis allow you to address these criticisms, we would be happy to look at a substantially revised manuscript. However, please bear in mind that we will be reluctant to approach the referees again in the absence of major revisions.

Response: Thank you very much for your important comments and nice suggestions! They are very valuable for improving our manuscript. We have studied these comments carefully and we are trying our best to solve these corrections which we hope meet with approval. All these corrections are as follows:

(1) The chiral-phase HPLC analysis of synthesized natural products (±)-asperazepanones A (**1**) and B (**2**) have been provided to certify the products in racemic forms and added in the supplementary information (Figures S43 and 44). And we analyzed all the synthesized intermediates (**10**, **7**, **11**, **4** and **12**) by chiral-phase HPLC. We confirmed that the intermediates **7**, **11**, **4** and **12** were racemic, and the chiral-phase HPLC chromatograms of them have been added in the supplementary information (Figures S39–42).

Chromatogram of the chiral-phase HPLC of synthesized compound (±)-1.

Compound (±)-1 was analyzed by HPLC using a Chiralpak IC chiral-phase column [5 μm , 4.6 \times 150 mm, hexane–ethanol eluent (70: 30), 0.6 mL/min].

Chromatogram of the chiral-phase HPLC of synthesized compound (±)-2.

Compound (±)-2 was analyzed by HPLC using a Chiralpak IC chiral-phase column [5 μm , 4.6 \times 150 mm, hexane–ethanol eluent (80: 20), 0.6 mL/min].

In the manuscript, we have added the sentence and the corresponding references about **7** as follows. Dieckmann cyclization of the optical amide **10** delivered the racemic tetramic acid **7** in 70% yield. Not surprisingly, although the Dieckmann cyclization strategy is a convenient way of preparing tetramic acids, it is accompanied by non-negligible epimerization at position C-5^{36,37}.

36. Peukert, S. et al. Design and structure-activity relationships of potent and selective inhibitors of undecaprenyl pyrophosphate synthase (UPPS): tetramic, tetronic acids and dihydropyridin-2-ones. *Bioorg. Med. Chem. Lett.* **18**, 1840–1844 (2008).

37. Poncet, J. et al. Tetramic Acid Chemistry. Part I. Reinvestigation of Racemisation during the Synthesis of Tetramic Acids *via* Dieckmann Cyclisation. *J. Chem. Soc., Perkin Trans. 1* 611–616 (1990)

(2) Optical rotation values of the optically active compounds **10**, (+)-**2** and (–)-**2** were $[\alpha]_{\text{D}}^{25} -2$ (*c* 0.6, MeOH), $[\alpha]_{\text{D}}^{25} +130$ (*c* 0.2, MeOH) and $[\alpha]_{\text{D}}^{25} -130$ (*c* 0.2, MeOH), respectively. Optical rotation values have been added in supplementary information.

(3) We have added the preparation of (-)-asperazepanone B (**2**) and experimental ECD spectra of (+)-**2** and (-)-**2** in supplementary information. (\pm)-Asperazepanone B (**2**) (10.0 mg) was resolved into the corresponding pure enantiomers (+)-**2** ($t_R = 58.0$ min, 4.0 mg), (-)-**2** ($t_R = 63.6$ min, 4.0 mg) by HPLC using a Chiralpak IC chiral-phase column [$5 \mu\text{m}$, 4.6×150 mm, hexane-ethanol eluent (80: 20), 0.6 mL/min]. (-)-Asperazepanone B [(-)-10aS-**2**]: yellow needle; $[\alpha]_D^{25} -130$ (c 0.2, MeOH).

(a) Chromatogram of the chiral-phase HPLC of synthesized compound (\pm)-**2**. (b) Chromatogram of the chiral-phase HPLC of (+)-**2**. (c) Chromatogram of the chiral-phase HPLC of (-)-**2**.

Experimental ECD spectra of (+)-**2** and (-)-**2**.

Reviewer #1:

The manuscript submitted by Shao and coworkers describes isolation, synthesis and biological evaluation of novel natural products, asperazepanones A and B. The authors isolated these natural products from *Aspergillus candidus*. The structures were determined by the standard NMR techniques and X-ray crystallography. They also revealed that asperazepanone A easily racemizes, while asperazepanone B is optically stable. The difference is attributed to the substitution of a methyl group on the nitrogen atom. The authors successfully synthesized these natural products, although the key Friedel-Crafts reaction proceeded only in low yield (compound 4 to 12, 22% yield). Biological evaluation of the natural products was also conducted, revealing that (+)-asperazepanone B showed inhibitory activity against nitric oxide (NO) production and inhibition on LPS-induced expression of TNF- α and IL-6.

The achievement in this manuscript will be informative for readers, because the authors disclosed chemistry and biology of natural products which have an unprecedented core, 6/7/5 diazatricyclic system. On the other hand, the manuscript has critical issues. This reviewer recommends publication of this work in this journal after the issues are properly addressed.

(1) preparation of (-)-asperazepanone B: Although the authors used (-)-asperazepanone B, the enantiomer of the natural product, in the biological evaluation, the method for preparation of (-)-asperazepanone B is not described in the manuscript and Supporting Information. It must be included.

Response: Thanks for your suggestions and comments on our manuscript. The method for preparation of (-)-asperazepanone B have been added in the manuscript and supplementary information.

In the manuscript, we have added the sentences “(±)-Asperazepanone B (**2**) was resolved into the corresponding pure enantiomers (+)-**2** and (-)-**2** by HPLC using a Chiralpak IC chiral-phase column.”

In the supplementary information, we have added the sentences “(±)-Asperazepanone B (**2**) (10.0 mg) was resolved into the corresponding pure enantiomers

(+)-**2** ($t_R = 58.0$ min, 4.0 mg), (-)-**2** ($t_R = 63.6$ min, 4.0 mg) by HPLC using a Chiralpak IC chiral-phase column [$5 \mu\text{m}$, 4.6×150 mm, hexane–ethanol eluent (80: 20), 0.6 mL/min]. (-)-Asperazepanone B [(-)-10a*S*-**2**]: yellow needle; $[\alpha]_D^{25} -130$ (c 0.2, MeOH).”

The synthesized compound (\pm)-2** was analyzed by HPLC using a Chiralpak IC chiral-phase column [$5 \mu\text{m}$, 4.6×150 mm, hexane–ethanol eluent (80: 20), 0.6 mL/min].**

(a) Chromatogram of the chiral-phase HPLC of synthesized compound (\pm)-**2**. (b) Chromatogram of the chiral-phase HPLC of (+)-**2**. (c) Chromatogram of the chiral-phase HPLC of (-)-**2**.

(2) For the synthesis of asperazepanones A and B, the authors used L-aspartic acid dimethyl ester as a substrate, but they mentioned in the manuscript that they synthesized natural products in racemic forms. Evidences that the products were racemic are not provided, and these should be added. In addition, this reviewer recommends the authors to add comments about when racemization occurred.

Response: Thanks for your suggestions and comments on our manuscript. The chiral-phase HPLC analysis of synthesized natural products (\pm)-asperazepanones A (**1**) and B (**2**) have been provided to certify the products in racemic forms and added in the supplementary information. And we analyzed all the synthesized intermediates (**10**, **7**,

11, 4 and 12) by chiral-phase HPLC. We confirmed that the intermediates **7, 11, 4** and **12** were racemic.

Chromatogram of the chiral-phase HPLC of synthesized compound (±)-1.

Compound (±)-1 was analyzed by HPLC using a Chiralpak IC chiral-phase column [5 μ m, 4.6 \times 150 mm, hexane–ethanol eluent (70: 30), 0.6 mL/min].

Chromatogram of the chiral-phase HPLC of synthesized compound (±)-2.

Compound (±)-2 was analyzed by HPLC using a Chiralpak IC chiral-phase column [5 μ m, 4.6 \times 150 mm, hexane–ethanol eluent (80: 20), 0.6 mL/min].

In the manuscript, we have added the sentence and the corresponding references about **7** as follows. Dieckmann cyclization of the optical amide **10** delivered the racemic tetramic acid **7** in 70% yield. Not surprisingly, although the Dieckmann cyclization strategy is a convenient way of preparing tetramic acids, it is accompanied by non-negligible epimerization at position C-5^{36,37}.

36. Peukert, S. et al. Design and structure-activity relationships of potent and selective inhibitors of undecaprenyl pyrophosphate synthase (UPPS): tetramic, tetrionic acids and dihydropyridin-2-ones. *Bioorg. Med. Chem. Lett.* **18**, 1840–1844 (2008).

37. Poncet, J. et al. Tetramic Acid Chemistry. Part I. Reinvestigation of Racemisation during the Synthesis of Tetramic Acids *via* Dieckmann Cyclisation. *J. Chem. Soc., Perkin Trans. 1* 611 (1990)

(3) It is recommended to add the optical rotation values ($[\alpha]_D$) for the optically active

compounds.

Response: Thank you for your reminding. Optical rotation values of the optically active compounds **10**, (+)-**2** and (-)-**2** were $[\alpha]^{29}_D -2$ (*c* 0.6, MeOH), $[\alpha]^{25}_D +130$ (*c* 0.2, MeOH) and $[\alpha]^{25}_D -130$ (*c* 0.2, MeOH), respectively, have been added in supplementary information.

(4) It is better that the structure of compound **3** appears in the manuscript.

Response: Thanks for your suggestions on our manuscript. We have added the structure of compound **3** in Figure 6.

(5) Supporting Information, page 24, line 334: $\text{CsCO}_3 \rightarrow \text{Cs}_2\text{CO}_3$

Response: Thanks for your reminding and suggestions. We have carefully revised numerous typographic errors in the manuscript.

Reviewer #2:

The manuscript Discovery, Total Syntheses and Potent Anti-inflammatory Activity of Asperazepanones A and B, Two Unique Pyrrolinone-Fused Benzoazepine Alkaloids from *Aspergillus candidus*” was submitted to Communications Chemistry for publication.

Broad comments:

The study describes the isolation of two novel benzoazepine alkaloids from *Aspergillus* cultures and the total synthesis of both isolated compounds. Moreover, some thoughts on the biosynthesis of the compounds are given and activity studies on NO production as well as on the expression of TNF- α and IL-6 are presented.

All the experiments are well described and the isolation and structure elucidation as well as the synthesis of the compounds is easy to follow. However, there are still some points to be addressed.

Figures: In contrast to the text, which is well written, some of the figures are misleading. In Figure 6, for example, which is showing the enamine-imino-tautomerism, in the starting molecule [(+)-**1**], the double bonds should move from the keto-oxygen towards the nitrogen. Therefore, the arrows should show in the other direction and one big arrow

should then show from the hydrogen to the keto-oxygen. Likewise, in the next molecule the arrow should not show towards the nitrogen but from the nitrogen and so on. Also, Figure 8 is misleading. Even though, it is a retrosynthetic analysis the fact that e.g. the arrow showing the decarboxylation is heading towards the carboxylated molecule appears somewhat strange. Maybe, a double pointed arrow would be more appropriate. Also for the other depicted reactions.

Response: Thank you for your valuable suggestions, they are very important for improving our researches and manuscript. We have studied these comments carefully and made the following efforts to improve the quality of our research and this manuscript. We hope these corrections meet with approval. All these corrections are as follows:

(1) In order to clarify Figure 6, we have re-summarized the literature on the proposed mechanism of the enamine-imino-tautomerism. For example, the study of “Enol-imino–Keto-enamine Tautomerism in a Diazepine Derivative: How Decisive Are the Intermolecular Interactions in the Equilibrium?” was published on *The Journal of Organic Chemistry* in 2019, authors investigated the diazepines tautomeric equilibrium between enol-imino, and keto-enamine, forms (see **Schemes 1 and 2**) by UV–vis and NMR spectroscopy in different solvents.

Scheme 1. Keto-enamine (O=DZP) and Enol-imino (OH-DZP) Forms of DZP

Scheme 2. RAHB Model Applied to the O=DZP Structure in the Solid State

Take another case, the study of “The stability and tautomerism of imines and

enamines based on benzothiazepines in protic solvents” was published on *Scientia Sinica Chimica* in 2014, the stability and tautomerism of imines and enamines in various protic solvents (methyl alcohol, ethyl alcohol, *tert*-butyl alcohol, acetic acid and CF₃CO₂D DMSO-*d*₆ solution) were investigated via ¹H NMR spectral data (Scheme 3).

Scheme 3. The reaction mechanism of imine to enamine was proposed.

On the basis of these related studies, we have redrawn Figure 6 and the racemization of (+)-**1** can be reasonably explained by a proposed mechanism. And the structure of compound **3** has been added. According to your advice, the arrow has been replaced with a double pointed arrow.

(2) Thank you very much for your comments and valuable suggestions on Figure 8. Since this step is very simple and straightforward, to avoid misunderstandings, we have removed the word “decarboxylation” and redrawn Figure 8.

Biological activity: The authors present effects of the compounds on NO production and on the LPS-induced expression of TNF- α and IL-6, claiming (+)-asperazepanone B as promising lead structure. However, no data on the cytotoxicity of the compound are given, which must not be neglected due to the known toxicity of *Aspergillus* species.

Response: Thanks for your suggestions and comments on our manuscript. We have tested the cell viability of (+)-**2** in RAW 264.7 cells, as can be seen in Figure S60, (+)-**2** showed no affecting cell viabilities at the concentration of 9 μM .

Specific comments:

Lines 58 and 82: The calculated m/z ratios of 1 and 2 should be 231.0770 and 245.0926, respectively.

Line 114: Please write “partially transformed”.

Line 158: Please write “Friedel-Crafts” here and in the whole manuscript.

Line 207 ff: Please indicate manufacturer and city and country of origin in the materials and methods section.

Response: Thanks for your reminding and suggestions. For the calculated m/z ratios of 1 and 2, the reviewer supplied with the [M+H] m/z ratios (231.0770 and 245.0926), not the [M+H]⁺ m/z ratios (231.0764 and 245.0921). Moreover, we have carefully revised numerous typographic errors in the manuscript. And we have added the

manufacturer and city and country of origin in the materials and methods section.

Special thanks for your valuable comments. We tried our best to improve our manuscript. All these corrections will not change the framework of this manuscript. Revised portions are marked in yellow color in the “Revised Manuscript” and “Supplementary Information” files.

Best wishes!

Sincerely,

Chang-Lun Shao

REVIEWERS' COMMENTS:

Reviewer #1 (Remarks to the Author):

The authors properly revised their manuscript according to the comments. This reviewer recommends publication of this work in communications chemistry.